# Detection of Complex Features of Car Body-in-White under Limited Number of Samples Using Self-Supervised Learning

**Chuang Liu, Kang Su, Long Yang \*, Jie Li and Jingbo Guo**

State Key Laboratory of Mechanical Behavior and System Safety of Traffic Engineering Structures, Shijiazhuang Tiedao University, Shijiazhuang 051130, China; liuc@stdu.cn.edu (C.L.); sukang2017@163.com (K.S.); lijie@stdu.edu.cn (J.L.); guojingbo66@163.com (J.G.)

\* Correspondence: yanglong233@163.com; Tel.: +86-136-2334-8742

**Abstract:** The measurement and monitoring of the dimensional characteristics of the body-in-white is an important part of the automobile manufacturing process. The process of using key point regression technology to perform online detection of complex features on body-in-white currently faces a bottleneck problem, namely limited training samples. Under the condition that the number of labeled normal map samples is limited, this paper proposes a framework for domain-independent self-supervised learning using a large number of original images. Under this framework, a self-supervised pre-order task is designed, which uses a large number of easily accessible unlabeled original images for characterization learning as well as a domain discriminator to conduct adversarial training on the feature extractor, so that the extracted representation is domain-independent. Finally, in the key point regression task of five different complex features, a series of comparative experiments were carried out between the method in this paper and benchmark methods such as supervised learning, conventional self-supervised learning, and domain-related self-supervised learning. The results show that the method proposed in this paper has achieved significant performance advantages. In the principal component analysis of extracting features, the representation extracted by the method in this paper does not show obvious domain information.

**Keywords:** body-in-white; complex features; detection; self-supervised learning; the training sample

## 1. Introduction

The rapid development of the automobile industry makes it occupy an important position in the global manufacturing industry and also makes automobile manufacturers put forward higher requirements for product quality. Accurately measuring and monitoring all kinds of dimensional features of an automobile body-in-white, especially complex features, is an important means to improve the quality control of automobile manufacturing. Figure 1 shows the comparison between simple features and complex features. Conventional 3D measurement technology is based on surface point cloud registration, which can better capture the surface information of complex features, but it is difficult to perform high-precision 3D reconstruction of these complex features [1–3]. This problem can be solved by using the keypoint regression method [4,5]. However, the core algorithm of this method is to regress keypoint coordinates from a two-dimensional normal map, and its training process requires a large number of normal maps with keypoint location labels. For industrial inspection and keypoint regression using surface normal maps, there is currently no database available in the industry. The acquisition of the normal map requires taking a large number of complex feature photos under multiple light sources and using photometric stereo technology to reconstruct the surface normal vector. Key point location calibration needs to go through relative position measurement of key points, camera calibration and coordinate transformation, which leads to a considerable cost to obtain the training data. Therefore, the question concerning how to complete the key point

regression task based on normal map under the condition of limited number of labeled samples has become a challenging topic in complex feature detection.

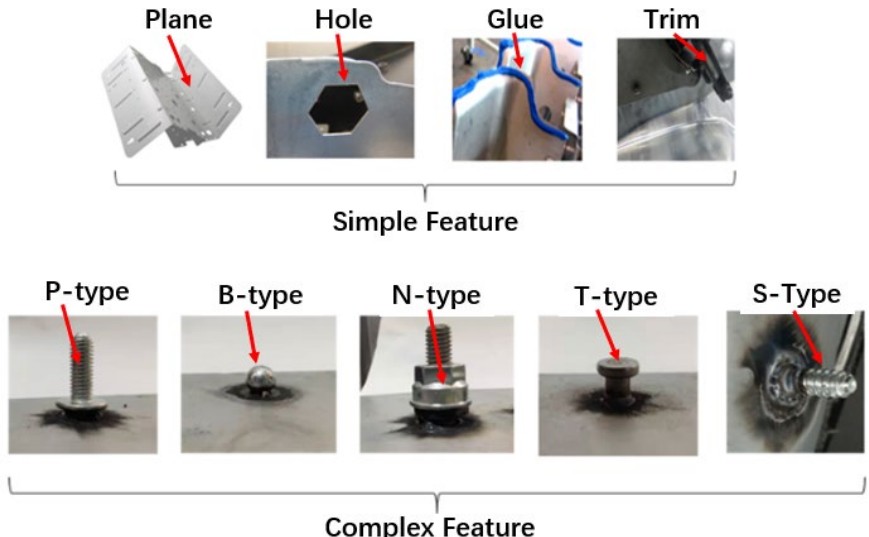

**Figure 1.** Comparison of simple features and complex features.

At present, the research on the key point regression problem is generally based on the deep learning technology of convolutional neural network, most of which are supervised learning [6]. The training process of supervised learning often requires a large amount of labeled data. In order to reduce the dependence of the keypoint regression method on the labeled training data, scholars try to use unlabeled data for self-supervised learning [7]. Inspired by self-supervised learning, Doersch et al. [8] proposed to learn high-level image representations by using predicting the relative positions of image patches as a pre-order task of self-supervised learning. This work has spawned research on patch-based visual representations for self-supervised learning, where Noroozi et al.'s model predicts the arrangement of "jigsaw puzzles" created from complete pictures [9]. Others generate well-designed image-level classification tasks in contrast to chunking-based methods. Caron et al. [10] used image clustering to create class labels, with classification tasks as pre-tasks. There are other pre-tasks for these methods such as image inpainting [11], image colorization [12], motion segmentation prediction [13] and spatial output tasks with high density. However, none of the above algorithms are applicable to the normal map-based keypoint regression problem.

Many scholars have studied the use of original images for self-supervised learning, and downstream tasks are also performed on the original images, but the representation of complex features under the original images is not obvious. The key point regression problem based on the original images is difficult to achieve satisfactory accuracy [14]. Research [15] shows that normal maps of complex features contain richer information than raw images. If self-supervision is performed in the original image and downstream tasks are performed in the normal map, the representation extracted by self-supervised learning will contain domain information, that is, the domain gap between the two images makes the representation learned by self-supervision in the poor performance in downstream tasks [16].

To address this challenge, this study proposed a highly data-efficient domain-invariant-self-supervised learning method (DISS). The model used by this method consists of two parts: a feature extractor and a regressor. Feature extraction used a jig-saw-based self-supervised pre-task (also translated as agent task) for training with both labeled and unlabeled data. In order to eliminate the domain distance between the original image and the surface normal map, this algorithm used a discriminator to assist training, and reduced the "KL" divergence between the probability distribution of the original image

and the surface normal map through adversarial training. The training of the regressor was performed entirely using labeled data.

This paper was divided into four parts. Part 1 was the introduction. Part 2 elaborated the framework of domain-independent self-supervised learning, including training data and problem description, self-supervised pretext-task training, network architecture, and training process. Part 3 was the experimental results and analysis, including the acquisition of data sets, complex feature key point regression experiments, and principal component analysis of extracted features. Part 4 was the conclusion.

## 2. Domain-Independent Self-Supervised Learning Framework

### 2.1. Training Data and Problem Description

Given a small amount of labeled normal map $D_t = \left\{ (x_i^t, y_i^t) \right\}_{i=1}^{m_t}$, in which $m_t$ represented the number of surface normal map, $x_i^t$ represented the surface normal map, and $y_i^t$ represented the corresponding label. Given a large amount of unlabeled raw image $D_s = \left\{ (x_i^s) \right\}_{i=1}^{m_s}$, in which $m_s$ represented the amount of raw image, $x_i^s$ represents the raw image. According to the assumptions of the question in this paper, there was $m_s \gg m_t$. Given a test set $D_v = \left\{ (x_i^v, y_i^v) \right\}_{i=1}^{m_v}$, in which $m_v$ represented the number of test data, $x_i^v$ represented the surface normal map, and $y_i^v$ represented the corresponding label. The goal of DISS was to use only $D_t$ and $D_s$ for training and to minimize the error on the test set $D_v$. The DISS training process was divided into two stages: training on the pre-task of the feature extractor, and training on the downstream task of the regressor.

### 2.2. Pre-Task Training

In self-supervised learning, pseudo-labels are usually obtained from unlabeled data to form pre-tasks. The design of the pre-task must meet two conditions: it can make the neural network converge, and at the same time, it must learn useful knowledge. Typical pre-tasks include: jig-saw puzzle, image rotation, image colorization, image completion, etc. In this problem, the jig-saw puzzle works best. Since the surface normal map was obtained from the original image through the photometric stereo algorithm, each normal map in the dataset $D_t$ corresponded to at least one original image, and these original images were formed into a dataset $D_r = \left\{ (x_i^r, y_i^r) \right\}_{i=1}^{m_t}$. Given training dataset $D_t$, $D_r$ and $D_s$, in which $x_i^s$, $x_i^r$ and $x_i^t$ were divided into $m_p$ jig-saw pieces. The nine pictures were randomly shuffled in order, and the shuffled different sequences were one-hot encoded to form pseudo-labels $y_i^{s*}$, $y_i^{r*}$ and $y_i^{t*}$. They formed new data sets $D_t^* = \left\{ (x_{i,j}^t, y_i^{*t})_{j=1}^{m_p} \right\}_{i=1}^{m_t}$, $D_r^* = \left\{ (x_{i,j}^r, y_i^{*r})_{j=1}^{m_p} \right\}_{i=1}^{m_t}$ and $D_s^* = \left\{ (x_{i,j}^s, y_i^{*s})_{j=1}^{m_p} \right\}_{i=1}^{m_s}$, which were used to train the pre-task. Figure 2 simplified the acquisition of the pre-order task using a four-piece puzzle. If the feature extractor can correctly distinguish the order of the puzzles, the features extracted by the extractor must also contain some kind of representation of the relationship between different parts of the object.

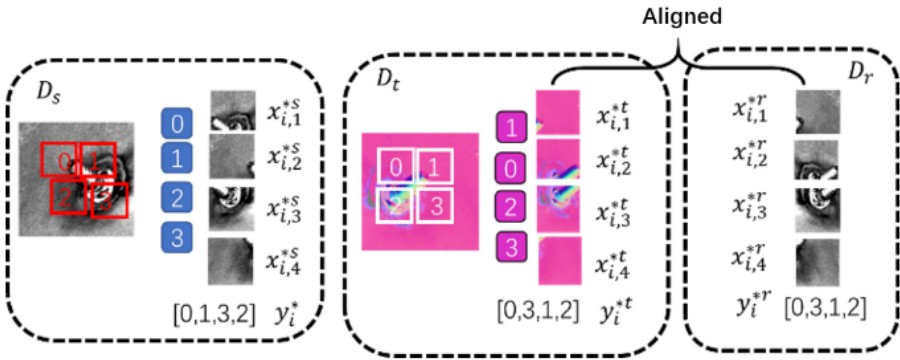

**Figure 2.** Schematic diagram of jigsaw pre-order task.

### 2.3. Network Architecture

The neural network architecture was shown in Figure 3. The network was divided into four parts: feature extractor E, sorter S, domain discriminator C and regressor R. As shown in the Figure 3, the training of DISS was divided into three stages. In the first stage of pre-task training, the features extracted by the feature extractor were sent to the sorter and the domain discriminator respectively. The target features extracted by the feature extractor had domain-independent variability through adversarial learning. During the regressor training process in the second stage, the parameters in the feature extractor were frozen, and only the parameters of the regressor were updated in the network. In the final fine-tuning stage, the parameters in the feature extractor and regressor were trained simultaneously.

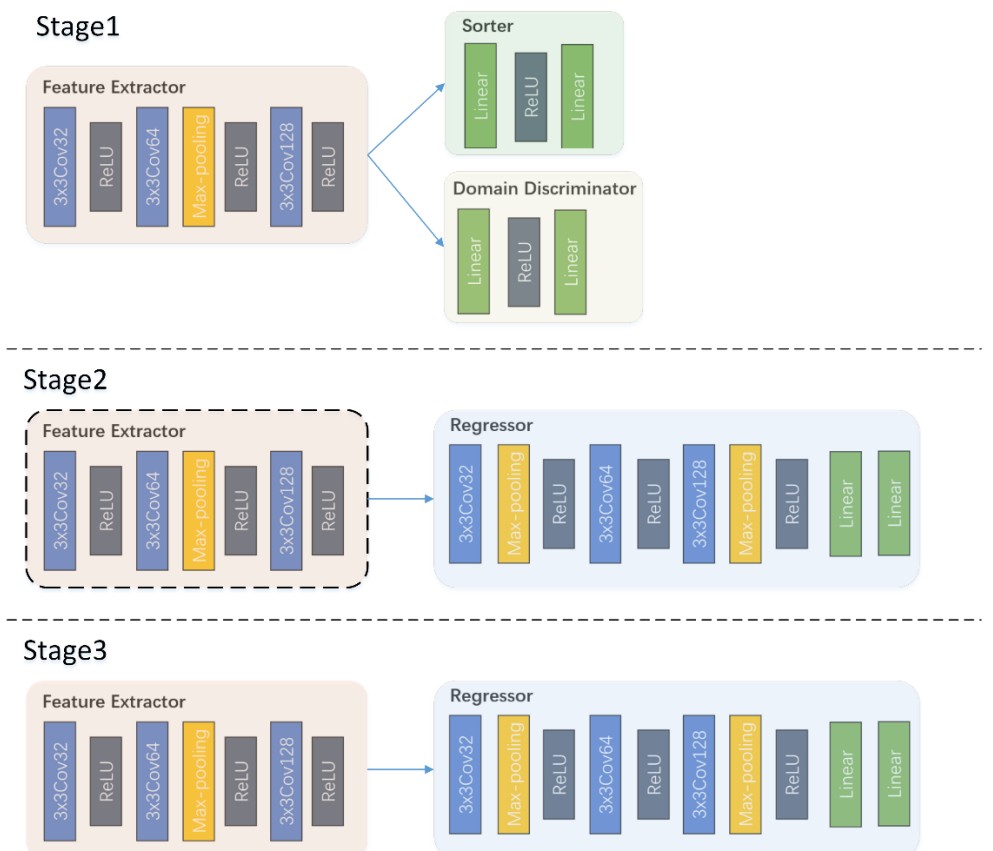

**Figure 3.** Network architecture.

The role of the feature extractor was to extract representations from the input that was useful for downstream tasks. The feature extractor consisted of 3 convolutional layers and a maxpooling layer, where the convolutional layers all used small convolution kernels.

The sorter was used for the pre-task. It sorted the features extracted by the feature extractor, and extracted the information of the object structure by forcing the feature extractor through learning. The sorter used a fully connected layer and a softmax layer as a classifier to predict the shuffled order of the jig-saw puzzle in the pre-task. The total number of categories of the pre-order task is equal to the factorial of the number of jig-saw puzzle pieces, which will lead to very many classification categories, and too many categories will cause training difficulties. Therefore, a small amount of all possible out-of-order combinations was usually extracted, and then one-hot encoding was performed. In this problem, 50 different out-of-order combinations were taken.

A large number of unlabeled samples in the training dataset came from the original image dataset, whose data distribution had a domain distance from the surface normal map dataset used for testing. The introduction of domain discriminator and adversarial

training, on the one hand, was used to discriminate whether the extracted features were from $D_t$ or $D_s$, on the other hand, it could reduce the domain distance between the features extracted from the original image domain and the surface normal map domain.

The regressor was used to estimate the complex feature keypoint locations through the extracted features, which consisted of three convolutional layers and one fully connected layer.

*2.4. Training Process*

The training process of DISS was divided into three stages: a pre-task stage, training regressor stage, and fine-tune stage.

In the pre-task stage, the data sets $D_t^* = \left\{ (x_{i,j}^t, y_i^{*t})_{j=1}^{m_p} \right\}_{i=1}^{m_t}$, $D_r^* = \left\{ (x_{i,j}^r, y_i^{*r})_{j=1}^{m_p} \right\}_{i=1}^{m_t}$ and $D_s^* = \left\{ (x_{i,j}^s, y_i^{*s})_{j=1}^{m_p} \right\}_{i=1}^{m_s}$ were used to train the feature extractor, domain discriminator and sorter, and the process was shown in Figure 4. To simplify the description, the example in Figure 4 was the case of a four-piece puzzle. The surface normal map $x_i^t$ was decomposed into four small pictures, and four one-dimensional feature vectors were obtained through the feature extractor E. The four vectors were shuffled in order $y_i^{*t}$ and concatenated into one dimension to form the representation:

$$f_i^{*t} = Cat(\left\{ E(x_{i,j}^t) \right\}_{j=1}^{m_p}) \tag{1}$$

where $Cat(\ )$ represented the vector concatenation operation. Did the same with $D_r^*$ and $D_s^*$ to get the representation:

$$f_i^{*s} = Cat(\left\{ E(x_{k,j}^s) \right\}_{j=1}^{m_p}), f_i^{*s} = Cat(\left\{ E(x_{k,j}^s) \right\}_{j=1}^{m_p}) \tag{2}$$

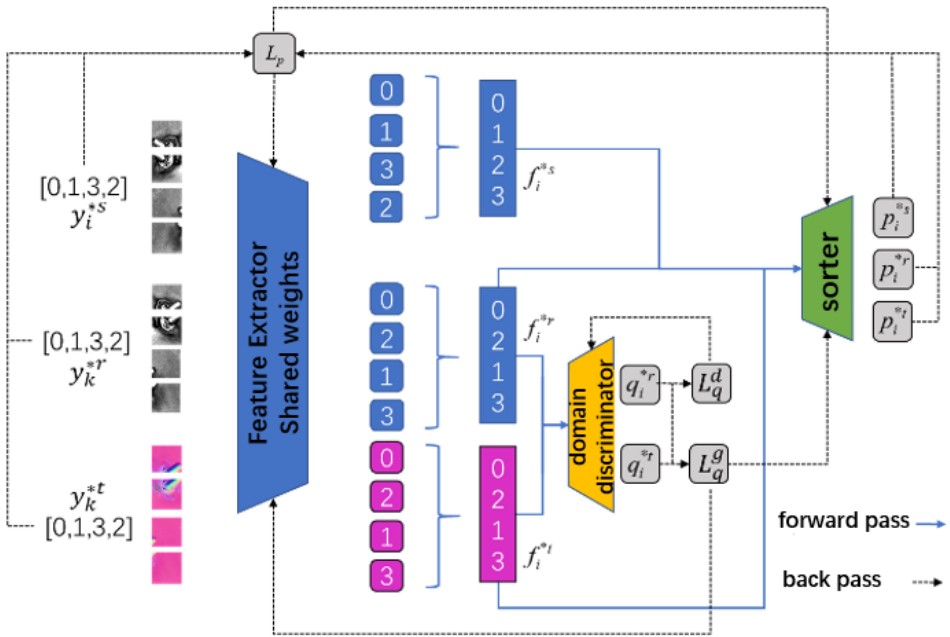

**Figure 4.** Data flow in the pre-task stage.

After the features were extracted, the sorter needed to sort them. The representation $f_i^{*t}$ was input to the sorter:

$$p_i^{*t} = S(f_i^{*t}), p_i^{*s} = S(f_i^{*s}), p_i^{*r} = S(f_i^{*r}) \tag{3}$$

The domain discriminator needed to determine which domain (original image or surface normal map) it came from, then the features were input into the domain discriminator to get:

$$q_i^{*r} = C(f_i^{*r}), q_i^{*t} = C(f_i^{*t}) \tag{4}$$

The ranking loss function $L_p$ was defined as:

$$L_p = \sum_{i=1}^{m_s} Ce(p_i^{*r}, y_i^{*r}) + Ce(p_i^{*t}, y_i^{*t}) + Ce(p_i^{*s}, y_i^{*s}) \tag{5}$$

where $Ce(\ )$ was the cross entropy loss function.

Defined the discriminative domain loss function $L_q^d$:

$$L_q^d = \sum_{i=1}^{m_s} -\log(1 - q_i^{*r}) - \log(q_i^{*t}) \tag{6}$$

Defined the extractor domain loss function $L_q^g$:

$$L_q^g = \sum_{i=1}^{m_s} -\log(1 - q_i^{*t}) \tag{7}$$

When training the pre-order task, the feature extractor and the sorter were trained simultaneously, and the domain discriminator was trained alternately with the other two neural networks. The pseudocode of an epoch training process was shown in Algorithm 1. Among them, $\lambda$ was the learning rate, $\theta_E$, $\theta_C$, and $\theta_S$ were the neural network parameters of the feature extractor, the domain discriminator, and the domain sorter, respectively.

---

**Algorithm 1** Pre-task training algorithm

---

Input: $D_t^* = \left[\left(x_{ij}^t, y_j^{*t}\right)_{j=1}^{m_p}\right]_{i=1}^{m_t}; D_r^* = \left[\left(x_{ij}^r, y_j^{*r}\right)_{j=1}^{m_p}\right]_{i=1}^{m_t}; D_s^* = \left[\left(x_{ij}^s, y_j^{*s}\right)_{j=1}^{m_p}\right]_{i=1}^{m_s}; \lambda$

  Initialize weights $\theta_E, \theta_C, \theta_S$
  for $i = 1 \rightarrow m_s$ do
    Extract features $f_i^{*t}, f_i^{*s}, f_i^{*r}$ according to formula (1) and (2)
    if $mod(i, 10) < 5$ then
      Calculate $q_i^{*r}, q_i^{*t}$ according to formula (4)
      Calculate the discriminative domain loss function $L_q^d$ according to formula (6)
      Calculate the extractor domain loss function $L_q^g$ according to formula (7)
      Update $\theta_C \leftarrow \theta_C - \lambda \Delta L_q^d$
    else
      Calculate the sort prediction $p_i^{*r}, p_j^{*t}, p_j^{*s}$ according to formula (3)
      Calculate the ranking loss function $L_p$ according to formula (5)
      Update $\theta_E \leftarrow \theta_E - \lambda \Delta(L_q^g + L_p)$
      Update $\theta_S \leftarrow \theta_S - \lambda \Delta L_p$
    end if
  end for

---

The training of downstream tasks was divided into two stages: training the regressor and fine-tuning the model.

Regression training used only the dataset $D_t = \left\{(x_i^t, y_i^t)\right\}_{i=1}^{m_t}$. The normal map in the dataset was input into the pre-task trained extractor to get the representation:

$$f_i = E(x_i^t) \tag{8}$$

The features were fed into the regressor to get the predicted coordinates of the keypoints:

$$\hat{y}_i = R(f_i) \tag{9}$$

The regression loss was calculated:

$$L_r = \sum_{i=1}^{m_t} MSE(\hat{y}_i, y_i^t) \tag{10}$$

Training process contained a total of $\delta$ epochs. The first $\delta_f$ epochs were the training regressor stage, which froze the model parameters of the feature extractor. After the $\delta_f$th epoch, the fine-tuning stage was entered, and the feature extractor and regressor were trained at the same time. The pseudocode of the training process was shown in Algorithm 2.

---

**Algorithm 2** Regression task training algorithm

---

Input: $D_t^* = [(x_i^t, y_i^t)]_{i=1}^{m_t}, \lambda_r, \tau, \delta, \delta_f$
Initialize weights $\theta_R, \theta_E$
for j = 1 → $\delta$ do
  for i = 1 → $m_t$ do
      Extract features $f_i^t$ according to formula (8)
      Extract features $\hat{y}_i$ according to formula (9)
      Calculate $L_r$
  end for
  if $j < \delta_f$ then
      Update $\theta_R \leftarrow \theta_R - \lambda_r \Delta L_r$
  else
      Update $\theta_R \leftarrow \theta_R - \lambda_r \Delta L_r$
      Update $\theta_E \leftarrow \theta_E - \lambda_r \Delta L_r$
  end if
  if $mod(j, \tau) == 0$ then
      $\lambda_r \leftarrow \lambda_r / 2$
  end if
end for

---

## 3. Experimental Results and Analysis

### 3.1. Acquisition of Datasets

In order to evaluate the performance of the DISS framework proposed in this study, a series of experiments were carried out in this paper. The experiment used the complex feature database proposed by Liu et al. [15], in which the labels of the original image data were completely hidden, and the labels of the normal map part were partly hidden, so as to simulate the situation of missing data labels. Figure 5 described the process of obtaining the four datasets used in this paper.

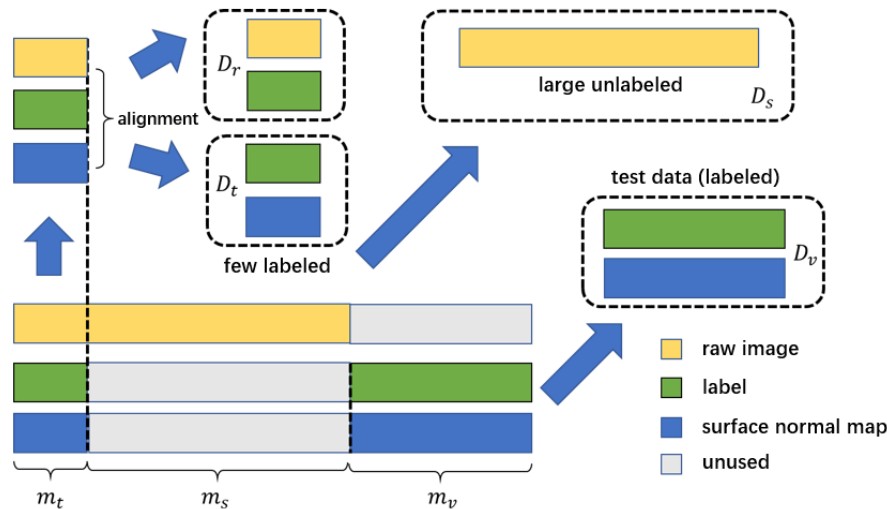

**Figure 5.** The datasets used in the experiment.

Liu's dataset contains raw images, surface normal maps, labels, and was referred to here as the original dataset. The $m_t$ sets of data containing the surface normal map and the label were taken from the original dataset to form the normal map with labels dataset $D_t$. The $m_r$ sets of data containing the original image and label were taken from this dataset to form the original image with labels dataset $D_r$. Note that the $D_t$ and $D_r$ were aligned, so they can use the same label data.

The $m_s$ sets of raw image data were taken from the original dataset to form the original image dataset $D_s$. According to the hypothesis of the research question, there was $m_t \ll m_s$. Finally, the $m_v$ sets of data containing the surface normal map and the label were taken from the original dataset to verify the accuracy of the algorithm.

### 3.2. Complex Feature Keypoint Regression Experiments

The methods proposed in this paper were compared with the benchmark methods SL (supervised learning), RSS (raw images self-supervised), and NRSS (normal maps and raw images self-supervised). The SL adopted the direct regression method, which only used labeled data for training. The RSS employed a Jig-saw-based self-supervised method, using unlabeled data for pre-task training. The NRSS method did not use adversarial training with a domain discriminator. The benchmark methods used for comparison were described in detail in Table 1.

**Table 1.** Comparison of different methods in the experiment.

| Method | Peculiarity | | | |
|--------|------------------------|-----------------------|-----------------|----------------------|
| | **Use the Original Image** | **Surface Normal Diagram** | **Self-Supervision** | **Confrontation Training** |
| SL | not | **Yes** | not | not |
| RSS | **Yes** | **Yes** | not | not |
| NRSS | **Yes** | **Yes** | **Yes** | not |
| DISS | **Yes** | **Yes** | **Yes** | **Yes** |

Here is a brief introduction to the difference between supervised learning and self-supervised learning. Training samples for supervised learning must be labeled sample data. Supervised learning is to train a model (learn a function) from a given labeled training data sets. When new test data is input, the result can be predicted according to the function. Self-supervised learning mainly uses proxy tasks to mine its own supervision information from large-scale unlabeled data, and trains the network through the constructed supervision information, so that it can learn valuable representations for downstream tasks.

Define the ratio of labeled data to unlabeled data,

$$\alpha = m_t / m_s \tag{11}$$

The ratio of unlabeled data to labeled data would affect how much the performance improvement brought by self-supervised method would be. This parameter was proposed for two reasons: First, it was to clarify the number of labeled data. Second, we could analyze the impact of changes in the number of labeled samples on DISS by changing the value of this parameter.

$\alpha = 5\%$ was adopted in Experiment 1, and the results were shown in Table 2.

In the experiments, the DISS proposed in this paper had the lowest average error. For the four complex features of N, P, S, and T types, DISS has the lowest error. For complex features of B type, the error of the DISS method was also very close to the lowest value. The experimental results fully reflected the advantages of DISS. Through self-supervised training, DISS was able to learn useful information about the geometric features of complex features in the pre-task using a large amount of unlabeled data, thereby gaining an advantage in the downstream task. Compared with RSS and NRSS, which also used self-supervised learning, DISS still had significant advantages. This was due to the fact that although RSS used self-supervised learning, the input data for the pre-task

and downstream tasks came from different domains, and this inter-domain bias resulted in that self-supervision could only improve the final performance to a limited extent. Comparing RSS and NRSS, it could be found that without using the domain discriminator and adversarial training, adding the normal map to training dataset did not improve the performance of the network on downstream tasks. This is due to the fact that although the discriminator was able to rank the input data from two different domains correctly in pre-tasks, there was no guarantee that the same representations were extracted from the input data from the two different domains. The structure information of complex feature learned by the feature extractor in the pre-task was mixed with different information from the two domains. This made it impossible to ensure that useful information can be extracted from input data from a single domain in downstream tasks.

**Table 2.** The average error of the proposed method and the three benchmark methods on five complex features in Experiment 1.

| The Characteristic Type | Feature Recognition | Average Error (in Pixels) | | | |
| --- | --- | --- | --- | --- | --- |
| | | DISS (Our Method) | SL | RSS | NRSS |
| N-type features | nut bolt | **9.22** | 11.88 | 10.82 | 11.12 |
| P-type features | flat bolt | **9.87** | 12.42 | 10.61 | 10.79 |
| S-type features | standard bolt | **8.23** | 9.33 | 8.88 | 8.94 |
| T-type features | t-bolt | **8.55** | 8.87 | 8.75 | 8.76 |
| Type B features | Ball-stud | 7.43 | **7.42** | 7.44 | 7.56 |
| average | - | **8.66** | 9.98 | 9.30 | 9.43 |

Compared with the SL method, DISS had more obvious advantages in N-type, P-type and S-type features. The reason was that these three kinds of complex features were relatively large in size and had rich texture and geometric features. In contrast, T-type and B-type features were smaller in size and had centrosymmetric shapes. In the pre-task, complex features with larger volume and richer surface information are easier to distinguish, while complex features with small volume and centrosymmetric shape are difficult to distinguish.

Experiment 2 was carried out when $\alpha$ takes three different values. Table 3 presented the results of Experiment 2. For the sake of brevity, only the average error of all kinds of complex features was shown here.

**Table 3.** The mean value of errors under different values of $\alpha$.

| $\alpha$ Value | Average Error (in Pixels) | | | |
| --- | --- | --- | --- | --- |
| | DISS (Text Method) | SL | RSS | NRSS |
| 5% | **8.66** | 9.98 | 9.30 | 9.43 |
| 10% | **7.69** | 9.07 | 8.23 | 8.20 |
| 20% | **5.88** | 6.38 | 6.40 | 6.48 |
| 40% | **4.52** | 4.68 | 6.62 | 6.59 |

The results of experiment 2 showed that as the amount of labeled data increased, the advantage of DISS over SL gradually decreased. This was due to the fact that the fourth stage of DISS had the same effect as the last few epochs of SL. When the number of labeled samples gradually increased, the effect of self-supervision was gradually masked by supervised learning.

### 3.3. Principal Component Analysis of Extracted Features

In order to further analyze the influence of DISS on the feature extractor, a principal component analysis (primary components analysis, PCA) experiment of extracted features was carried out in this paper. The images $x_i$ in one dataset were imported into the feature extractor to get the representation of all samples, which were then expanded into

n-dimensional vectors $\vec{f}_i' \in \mathbb{R}^{n \times 1}$, and all of the extracted features were represented as a matrix:

$$F' = [f_1', f_2', \ldots, f_m'] \tag{12}$$

To normalize the data:

$$f_{i,j} = \frac{f'_{i,j} - \overline{f}_i'}{\sqrt{s_{i,i}}}, i = 1, 2, \ldots, n, j = 1, 2, \ldots, m \tag{13}$$

In which $f_{i,j}$ represented the elements in the normalized matrix **F**. To calculate its sample correlation matrix as:

$$R = \frac{1}{n-1} FF^T \tag{14}$$

The eigenvalues and eigenvectors of R were calculated to acquire two eigenvectors corresponding to the largest two eigenvalues. The matrix combined by the two eigenvectors was multiplied by the original normalized matrix:

$$\begin{bmatrix} \widetilde{x} \\ \widetilde{y} \end{bmatrix} = \begin{bmatrix} \alpha_1 \\ \alpha_2 \end{bmatrix} F \tag{15}$$

In which $\widetilde{x}, \widetilde{y} \in \mathbb{R}^{1 \times m}$ were the abscissa and ordinate of all dimensionality-reduced data, respectively. The original image and the surface normal map in the dataset were input into the feature extractor separately to get:

$$\vec{f}_i^s, \vec{f}_i^r \in \mathbb{R}^{n \times 1}, i = 1, 2, \ldots, m_p \tag{16}$$

The two vectors were stitched into a matrix:

$$F^a = [\vec{f}_1^s, \vec{f}_2^s, \ldots, \vec{f}_i^s, \vec{f}_1^r, \vec{f}_2^r, \ldots, \vec{f}_i^r] \in \mathbb{R}^{n \times 2m_p}, i = 1, 2, \ldots, m_p \tag{17}$$

Principal component analysis was performed on $F^a$ according to the above method to obtain dimensionality-reduced representations from different domains. Figure 6 showed the results for different kinds of complex features.

It was easy to find from the experimental results that the features from the normal map extracted by the DISS method are more indistinguishable from the features from the original image. On the contrary, the features extracted by RSS and NRSS methods carried obvious domain information. For example, the features from the normal map extracted by the RSS and NRSS methods were clustered together, compared with the features from the original image were more divergent. The features extracted by the DISS method did not have this characteristic. Besides, the features extracted from the normal map by the RSS and NRSS methods had a smaller abscissa, while the features of the original image had a larger abscissa. This pattern was not obvious in the features extracted by the DISS method. Overall, the feature distributions extracted by the DISS method for inputs from the two domains were more similar. In contrast, the features extracted by RSS and NRSS carry obvious domain information. In the training of the pre-task, the amount of original image data was much larger than the amount of normal map data. How to effectively utilize the features extracted from the original image was the key issue of self-supervised learning in this research. The features extracted by the DISS method did not carry obvious domain information, which enabled the features learned from raw images to be used in downstream tasks using normal maps as well. The features extracted by the RSS and NRSS methods had obvious domain information, and the distribution of the extracted features in the two domains had obvious distance, which made the features learned from the original image difficult to be used by downstream tasks.

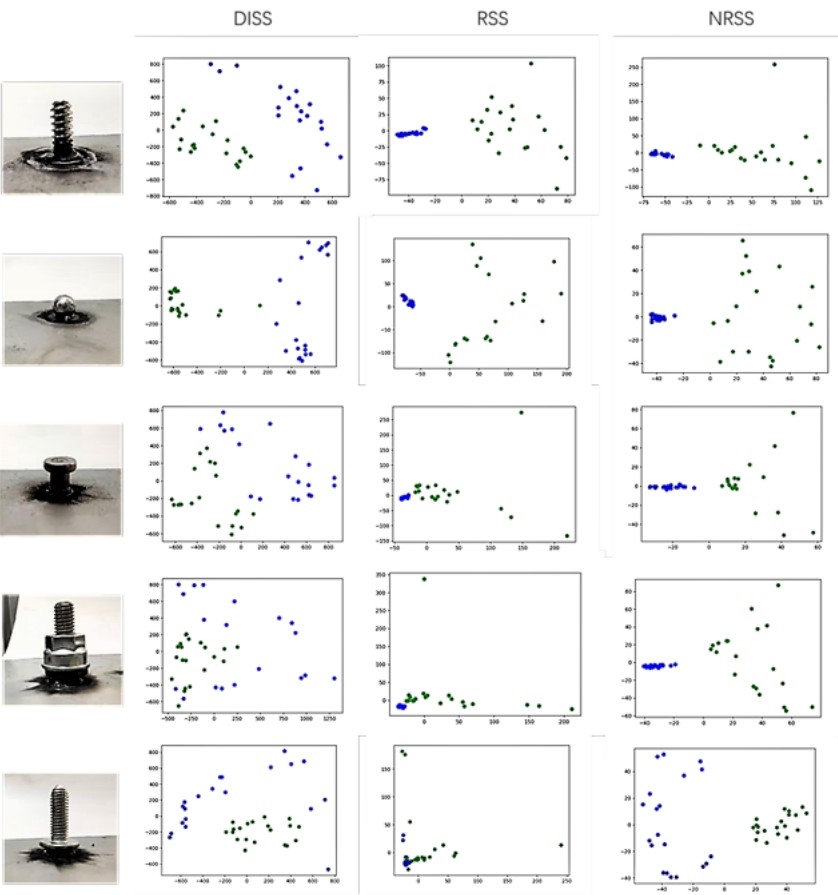

**Figure 6.** Principal component analysis of extracted features from two inputs.

## 4. Conclusions

Aiming at the limited training samples of the complex feature detection technology of body-in-white, the few-shot learning technique based on solving the domain gap problem in this paper can effectively reduce the dependence of the complex feature measurement system on the labeled normal map. The proposed self-supervised learning and domain-adaptive learning method had 1.32 pixels lower error than the supervised learning method using normal maps, 0.64 pixels lower than the self-supervised learning method using the original image, and 0.51 pixels lower than the self-supervised learning method using normal maps. In extracting feature principal component analysis, the representations extracted by our method do not exhibit obvious domain information, which proves the effectiveness of our method. The results of this paper are of great significance for improving the data efficiency of complex feature detection methods and reducing implementation costs.

Detection of fusion of complex features and simple features is the direction of further research. There may be simple features with complex features in areas other than automobile body-in-white dimension control. For example, it only has complex geometry, but no complex reflection properties. How to use this characteristic to develop a higher performance and lower cost detection system is a topic worthy of further study.

**Author Contributions:** Conceptualization, C.L.; methodology, J.L. and J.G.; software, K.S.; validation, C.L.; formal analysis, C.L.; investigation, C.L.; resources, J.L. and J.G.; data curation, K.S.; writ-ing—original draft preparation, C.L.; writing—review and editing, L.Y.; visualization, C.L.; su-pervision, J.G.; project administration, C.L.; funding acquisition, C.L. All authors have read and agreed to the published version of the manuscript.

**Funding:** This research was funded by State Key Laboratory of Mechanical Behavior and System Safety of Traffic Engineering Structures Independent Project, grant number ZZ2020-37 (Chuang Liu), the Science Research Project of the Education Department of Hebei Province, Grant No. ZD2021093 (Kang Su), National key Research and development Program: 2020YFB1709502 (Jie Li) and National key Research and development Program: 2020YFB1709504 (Jingbo Guo).

**Institutional Review Board Statement:** Not applicable.

**Informed Consent Statement:** Not applicable.

**Data Availability Statement:** Not applicable.

**Acknowledgments:** The authors thank Northeastern University Machine Vision Laboratory for technical support and help in this research, and the anonymous reviewers and copy editor for valuable comments proposed.

**Conflicts of Interest:** The authors declare no conflict of interest.

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
