# Peer review of "Detection of Complex Features of Car Body-in-White under Limited Number of Samples Using Self-Supervised Learning"

_coatings, doi:10.3390/coatings12050614_

Round 1

Reviewer 1 Report

coatings-1665929-peer-review-v1

The authors presented a novel method based on deep learning to implement self-learning of body-in-white of cars’ images. The manuscript still needs some modifications to get accepted for publication. Moderate English revision should be conducted before the manuscript is resubmitted.

The title should be modified to include the novel method that has been presented in the study such as “Detection of complex features of car body-in-white under limited number of samples using self-supervised learning         

INTRODUCTION

 Figure 1 : Simple features- the feature (hole, glue, etc., ) needs to be referred to by an arrow.

Complex features: The feature in each image needs to be referred to by an arrow.  

Line 69:  “So the key point regression”? Delete “so”.

Line 78: (DISS) not DISS.

Line 83: Identify “KL”.

Figure 7. “The datasets…” not “The dataset…”

Line 208: Delete the first “and”

Lines 219-221: What type of supervised learning methods were used? This should be stated.

The authors should explain the supervised learning in section 2

Result and Discussion

Table 2. The authors should identify “the characteristics types“, N-type features, …etc.,

The authors need to specify how the models were validated. Were the results for the test set?

The authors need to state the meaning of alpha and why it was used.

The authors need to put more effort into the discussion section and compare their results with previous studies' results.

Reviewer 2 Report

The paper is well written and structured, it presents important research, but the experimental results can be improved to test the robustness of the algorithm, with datasets of real scenarios from industrial images or in live process from a camera.

At the end of the introduction, the organization of the paper must be written.

Figure 3 must be redrawn because the interconnections between the blocks of the architecture is not in a most explicit way.

Fit the dimensions of table 1 to the model.

Experimental results can be improved, in particular by testing the algorithm in a real-time process, where images are acquired and processed from a camera.

The direction of future work needs to be identified.
